# MooseNet: A Trainable Metric for Synthesized Speech with a PLDA Module

*Ondřej Plátek, Ondřej Dušek*

Charles University, UFAL, Prague, Czech Republic

{oplatek,odusek}@ufal.mff.cuni.cz

## Abstract

We present MooseNet, a trainable speech metric that predicts the listeners' Mean Opinion Score (MOS). We propose a novel approach where the Probabilistic Linear Discriminative Analysis (PLDA) generative model is used on top of an embedding obtained from a self-supervised learning (SSL) neural network (NN) model. We show that PLDA works well with a non-finetuned SSL model when trained only on 136 utterances (ca. one minute training time) and that PLDA consistently improves various neural MOS prediction models, even state-of-the-art models with task-specific fine-tuning. Our ablation study shows PLDA training superiority over SSL model fine-tuning in a low-resource scenario. We also improve SSL model fine-tuning using a convenient optimizer choice and additional contrastive and multi-task training objectives. The fine-tuned MooseNet NN with the PLDA module achieves the best results, surpassing the SSL baseline on the VoiceMOS Challenge data.

**Index Terms**: evaluation, metric learning, mean opinion score prediction, speech synthesis

## 1. Introduction

We present the MooseNet metric, which predicts the Mean Opinion Score (MOS) from a single utterance of synthesized speech. Using MOS from recruited listeners is a well-established standard for evaluating text-to-speech (TTS) and voice conversion (VC) systems [1], and MOS prediction metrics [2] are a way to automate this process. The organizers of the 2022 VoiceMOS Challenge [3] released a large dataset with MOS annotations for TTS and VC systems' outputs (called BVCC), so that MOS prediction metrics can be trained in a supervised manner. One of the aims of the VoiceMOS challenge was investigating the use of self-supervised learning (SSL) speech models [4, 5] finetuned for the MOS prediction task. Using SSL models requires fewer annotated utterances than training NN models from scratch, but fine-tuning on a limited number of examples may lead to overfitting to the audio channel and speech properties of the training data, hurting performance on non-matching examples. To investigate this, the VoiceMOS challenge included two tracks: The main track with 4,974 training utterances and the Out-of-Domain (OOD) training set with only 136 utterances, intended to evaluate the applicability of MOS predictors trained on the main track to a new domain.

While the VoiceMOS main track data proved to be large enough for building a robust SSL-based MOS predictor and SSL-based models are the current state of the art on the task [6, 7, 2, 8], it is still unclear how many MOS annotated utterances are really needed for finetuning an SSL model. By experimenting on both VoiceMOS main and ODD track datasets, we investigate if pretraining on a larger dataset is crucial for

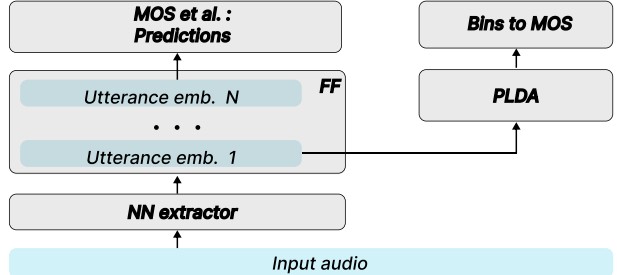

Figure 1: *PLDA can use any layer after global pooling as utterance level embedding as its features.*

fine-tuning the MOS predictor to a new small dataset and how much data is needed for it. We show in a simple ablation study that even with 5% training data, SSL fine-tuning outperforms the previous non-SSL state-of-the-art LDNet model [2]. In addition, we present an even more effective low-resource alternative approach to the traditional finetuning paradigm of SSL models by reframing the MOS-prediction regression as classification and introducing Probabilistic Linear Discriminative Analysis (PLDA). In contrast to previous systems, PLDA performs very well even for a few hundred annotated utterances. Furthermore, its projections are computed very fast on a single CPU and thus require minimal resources for training and inference. Importantly, PLDA can be easily combined with existing neural network models.

Our contributions are the following:

(1) We introduce a new SSL-based neural network MOS prediction model, dubbed MooseNet, which is based on models of Cooper et al. [6] and Saeki et al. [7] and further improves model training, optimizing hyperparameters and introducing multi-task learning. The MooseNet neural network reaches near state-of-the-art performance on the VoiceMOS data.

(2) We introduce PLDA as a convenient method for adapting pre-trained models to downstream tasks. We demonstrate the use of PLDA on several variants of SSL models [4, 5].

(3) In ablation studies on VoiceMOS data, we investigate the performance of PLDA and several strong neural baselines based on the amount of available data. We show that PLDA consistently improves SSL models, matching state of the art on VoiceMOS. Models without finetuning to the MOS prediction task as well as specifically fine-tuned models benefit from using PLDA.

(4) We release our implementation, experimental setup, pre-trained models, and system outputs to ease future research.[1]

---

[1] https://github.com/oplatek/moosenet-plda

## 2. Related Work

**Intrusive Metrics:** Synthesized speech is hard to evaluate automatically. Early works were inspired by de-noising audio evaluation, which compares clean reference audio signal to noisy input or denoised system output signal. The so-called intrusive metrics MCD [9] and STOI [10] reported a moderate correlation with human judgment. Our approach follows the most recent **non-intrusive** metrics and does not need any references (see below). Still, we experimented with STOI prediction on synthetic data as an additional criterion for multi-task learning (see §3).

**Frechet Audio Distance:** The first step towards trainable metrics was Frechet Audio Distance (FAD), which showed promising results for music recording evaluation [11]. FAD measures the distance between a set of reference recordings and an unpaired set of hypotheses using features from a music classifier. Binkowski et al. [12] extended FAD for evaluating synthesized speech using features from the DeepSpeech2 speech recognition model [13]. Similarly to FAD, PLDA uses an existing neural model for its features. However, PLDA is a trainable model and can be fine-tuned to the task on top of the neural features. Note also that the self-supervised learning (SSL) approach proved to be superior to purely ASR-trained models like DeepSpeech2 for transfer learning to MOS prediction [4, 5].

**Trainable MOS Predictors:** Trainable neural metrics directly predicting MOS dominate in automatic speech evaluation as they show a high correlation with human evaluation, not only on the system level but also on the utterance level [6]. Furthermore, they generalize well to diverse speech properties such as tempo variation or different prosody when trained on enough data. The MOSNet metric became the first widely adopted trainable neural network metric for synthesized speech that does not use references [14]. The MOSNet neural network was trained from scratch on the Voice Conversion Challenge (VCC) 2016 and 2018 datasets [15], unlike SSL-based methods described below.

**SSL-based MOS Predictors:** Strong self-supervised learning pre-trained models [4, 5] improved many downstream tasks when fine-tuned on them including speech MOS prediction. SSL-based MOS predictors are currently state-of-the-art for the task [3, 7, 8].[2] The organizers of the VoiceMOS challenge released a simple yet well-performing SSL baseline (B01 in [3]) based on the Wav2vec model, which outperforms MOSNet and other prominent systems by a large margin. The SSL baseline uses the architecture of the SSL encoder to obtain frame-level representation, a global pooling to obtain utterance-level embeddings and adds a feed-forward (FF) neural network to perform regression for MOS predictions. The baseline initializes the encoder weights from a checkpoint and fine-tunes all parameters of NN jointly. See §3 for the MooseNet architecture and training description, which is inspired by UTMOS [7], the winning VoiceMOS system which is based on the organizers' baseline.

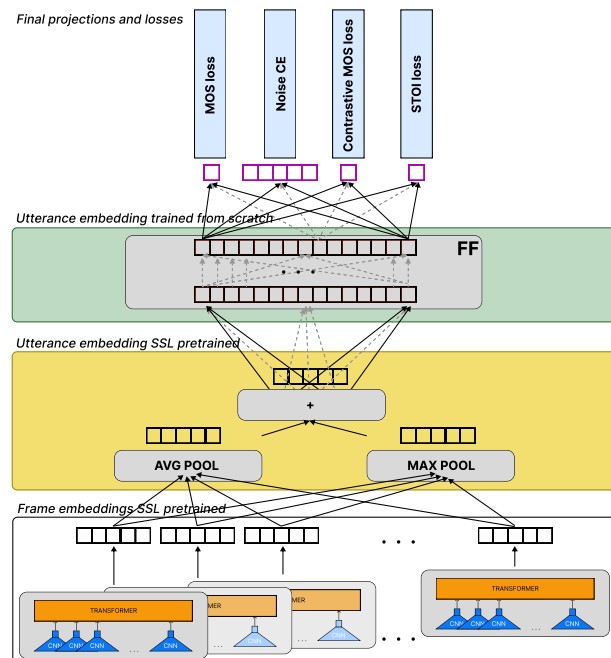

Figure 2: *MooseNet architecture is based on pre-trained SSL models. Frame-level embeddings are transformed to utterance level by global pooling. FF layers and final projections are the only parameters trained from scratch.*

## 3. MooseNet Neural Network

**Network Architecture:** Following the SSL VoiceMOS challenge baseline [6], the network architecture of MooseNet consists of four main blocks depicted in Figure 2. The SSL encoder (we use Wav2vec [4] or XLSR [5]) extracts audio-frame-level features. The max and average global pooling operations project the $T$ frame-level vectors $v_{t \in 1,...,T}$ to two fixed-size vectors representing the whole utterance.[3] We sum the vectors together before passing them to a simple feed-forward (FF) NN. The last layer consists of projections to multiple single scalars for regression and a single vector for classification. We always initialize the encoder from its SSL pre-trained checkpoint. Only the parameters of the FF network and the final linear projections are initialized randomly and trained from scratch. As a result, only a negligible proportion of parameters are trained from scratch.[4]

**Improvements to the Training:** We switch to using LAMB optimizer [16] with Noam scheduler [17] and fine-tune the needed warm-up steps and learning rate on development data. Inspired by Saeki et al. [7], we adapted contrastive loss to boost ranking performance. We hypothesize that contrastive loss performs well since contrastive training is known to be less sensitive to overfitting than cross-entropy loss or regression. [7] Following Lakshminarayanan et al. [18], we also observe that using Gauss loss stabilizes and improves the training which we

---

[2]Consistently with literature, we found out training MooseNet with weight initialization from a SSL model is superior to the random initialization.

[3]Compared to the VoiceMOS baseline, we added the max operation in addition to the average operation.

[4]For *Wa2vec-small* the ratio is 0.001, and for *XLSR* it is 0.0003.

explain as better modeling the MOS variance.

**Data Augmentation and Multi-Task Training:** We use audio data augmentation to generate data for multi-task training – we create degraded versions of the MOS-annotated synthesized speech data, mixing clean utterance together with noise from a database with a known signal-to-noise ratio (SNR). We then use the original-degraded pairs of synthesized speech in a multi-task training setting with four different tasks, depicted in blue in Figure 2, in addition to the Gauss and contrastive losses described above: We train the model to predict (1) STOI [10] and (2) MCD [9] objective metrics [19] where we use the original synthesized speech as the reference and its noise-degraded version as the hypothesis.[5] We also use (3) a regression task to predict the SNR which was used during the noise augmentation. Finally, (4) the noise classification task predicts the type of noise which was used for data augmentation, given the original and degraded synthesized speech.

**Compatible Methods:** The UTMOS system focused on achieving the best performance possible and combined many methods. In our MooseNet NN implementation, we focus on approaches that do not modify neural network architecture and do not require additional annotation except for MOS for each utterance. We avoid Listener Dependent (LD) modeling despite multiple works reporting its benefits [7, 8, 20] because we can achieve similar gains simply by improving the fine-tuning training procedure described in §3. Ensembles [21, 18] and larger SSL models like XLSR [22] or Hubert Large [23] are obvious approaches how to improve system performance but are computationally demanding. Both methods are compatible with our approach, but unlike Saeki et al.[7] we do not evaluate them in our paper.

# 4. PLDA for MOS Prediction

PLDA is a classification generative probabilistic model, well-known in face recognition [24] and speaker verification [25] for its robust likelihood estimates. The PLDA in speaker verification uses a fixed-size vector embedding of an utterance as its features [25] and inspired us to evaluate PLDA for MOS predictions.

The PLDA estimates mean and covariance matrixes for its Gaussian Mixture Model (GMM) and LDA projection matrix to its latent space.[6] The GMM model is used at inference to compute posterior probabilities for each class given the input. The PLDA training minimizes intra-class variance and maximizes the inter-class variance of the GMM model.

For PLDA, we frame MOS prediction as a classification task from audio into bins representing the possible range of MOS between 1 and 5. The bin boundaries are estimated on training data to distribute MOS score samples in bins equally. Note that we apply PCA to decorrelate the neural embedding features before we use PLDA.

After training the PCA and PLDA, we keep the estimated matrixes for inference. At inference, we project the NN embedding using the PCA and PLDA matrixes to obtain the parameters of the GMM model which predicts the posterior probability for each class – a MOS bin. The final PLDA MOS score is computed as a weighted sum of MOS-bin-centers values and the posterior probability of each bin.

# 5. Experiments

We evaluate our experiments on VoiceMOS data using the recommended system-level Spearman Rank Correlation Coefficient (SRCC) and Mean Squared Error (MSE) metrics.[7] Both compare the system predictions to human-obtained scores: MSE computes the mean distance between the predicted and human scores, while SRCC compares how the predictor maintains ranking with respect to human scores.

We conducted all preliminary experiments and model selection on the development set and used the test set only to evaluate experiments presented in Table 1. We run each experiment with ten different random seeds and report the mean and standard deviation from these ten runs. This improves on the VoiceMOS challenge practice, where single system runs are reported and the random seed effect is unknown.

**Overall Experiment Plan:** We train and evaluate our models on the VoiceMOS main track and OOD track training and test sets. By default, we train one model on the main track and evaluate it on the main track test set. We then further finetune this model on the OOD track and evaluate the result on the OOD test set. However, we run more experiments summarized below and in Figure 3, answering the following research questions:[8]

(RQ1) How do the individual MooseNet NN training methods help the fine-tuning performance on the main track? Compare the first six experiments in Table 1.

(RQ2) How does the MooseNet NN perform on the OOD track when fine-tuned only on the main track? See *OOD: W2V_main* experiment.[9]

(RQ3) Does PLDA perform better when used on top of MooseNet fine-tuned for a given dataset? Compare *OOD: W2V_main+PLDA_ood* and *OOD: W2V_ood+PLDA_ood*.

(RQ4) How much does the performance drop if PLDA uses an unmodified Wav2Vec model? See the *Main: W2V+PLDA_main*, *Main: W2V_main+PLDA_main*, *OOD: W2V+PLDA_main*, *OOD: W2V+PLDA_ood* experiments.

(RQ5) How does the performance of MooseNet NN change for a reduced number of data? See ablation study experiments W2V_main-*X* where X is $50\%, 5\%, 136$ where 136 examples correspond circa to 2.7% of the training data.

(RQ6) How does the performance change if we use a larger XLSR model instead of a Wav2vec small model? Find all experiments with *XLSR* and compare the same experiments with the *W2V* variant.

**Data Preparation:** We augmented the input audio data during training using time-domain volume and tempo augmentation. The Lhotse volume augmentation is used with 0.8 probability, and the scaled factor was sampled from range $[0.5, 2]$. The tempo perturbation factor is sampled from the $[0.9, 1.08]$ range. We filter out utterances shorter than 1s and longer than

---

[5] Note that in this way, we can use STOI prediction in a non-intrusive way as human reference outputs are not required for the setup.

[6] See Ioffe [24, §3.2] for PLDA parameters learning details.

[7] The KTAU and PCC metrics can be safely omitted because they are highly correlated with SRCC [3].

[8] The experiments for ablation studies regarding (RQ1) and (RQ5) were not included in the figure because they share the same overall structure as the *W2V_main* experiment.

[9] The *OOD: W2V_main* experiment name denotes that the MooseNet NN was initialized with the W2V checkpoint and further fine-tuned on the *main* track but evaluated on the *OOD* test set. Other experiments are labeled analogically.

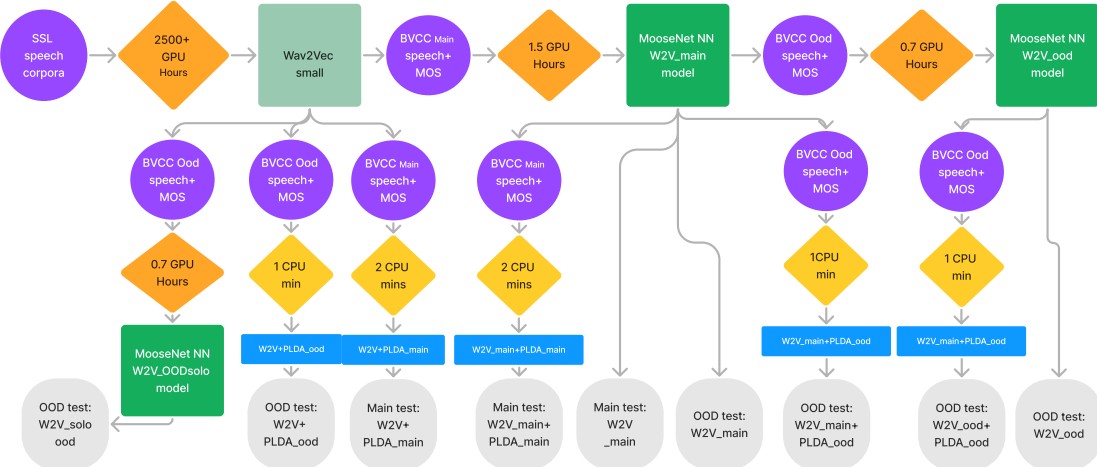

Figure 3: *Training workflow of the Experiments using the MooseNet NN checkpoints (dark green rectangles), Wav2vec checkpoint (a light green rectangle), or additional PLDA parameters (light blue rectangles). The purple circles describe the data used for the training, and the orange rhombuses show the time of CPU or GPU used for the training. The bottom row of rounded rectangles shows the names of the experiments including if the Main or OOD test set was used.*

12s for training. We use bucketing with 20 buckets, and we let our experiments stop early based on validation SRCCC with the patience of 30 epochs.

**Batching:** The number of utterances in a batch differs according to the bucketing strategy and random sampling, but it cannot exceed 80s of audio. Half of them are clean utterances, and the other half are noise-degraded variants as described in §3. The utterances are degraded by mixing the clean audio with noises from the MUSAN [26] dataset with the SNR sampled from $[10, 20]$ dB interval for each utterance. We use the Lhotse [27] library for the mixing, bucketing utterances into 20 buckets, and sampling the utterances.

**MooseNet NN Fine-Tuning:** Our best-performing NN models follow the training structure of a neural network SSL baseline [6]. We first fine-tuned a pre-trained SSL model [4] [5] on the main track. See *W2V_main* and *W2V_main-XLSR*. Later, we use the pre-trained checkpoints on the main track for fine-tuning the models on the OOD track. See *W2V_ood* and *W2V_ood-XLSR* experiments. We also tried to fine-tune the Wav2vec model directly on the OOD track in *W2V_oodsolo* experiment which performed surprisingly well. In our second NN non-standard experiment, we evaluated the MooseNet NN trained only on the main track on the OOD test set which produced poor results as expected.

**MooseNet NN Fine-Tuning Hyperparameters:** By using Noam LR scheduler with 1500 warmup steps and LAMB optimizer with a learning rate of 0.001 and weight decay of 0.0001, we train faster, achieve better results for the NN model, and we

can utilize a larger pre-trained XLSR [22] model.[10] Following Leng et al. [28], we clipped the logCosh and Gauss losses.

Based on informal experiments, we observed that using only one hidden layer on top of global pooling is enough. We also verified that reducing the hidden embedding size to 32 is not inferior to a larger embedding size. The smaller embedding size proved beneficial to train the PLDA backend.

**PLDA Model Variants:** We train PLDA in several setups. In two experiments *W2V_main+PLDA_main* and *W2V_ood+PLDA_ood*, we evaluate PLDA on OOD and main test set and train it also on the main and test sets respectively using fine-tuned NN model for the given set. We also investigated how PLDA can perform when it uses a non-finetuned model for given datasets because PLDA can be trained much faster than the NN model. See *W2V_main+PLDA_ood* and *XLSR_main+PLDA_ood* experiments. For the above mention experiments, we used an NN embedding size of 32, we discretized MOS scores to 32 bins, and we let PCA and PLDA algorithms keep all found dimensions of their projections. Finally, we evaluated PLDA performance on unmodified Wav2vec and XLSR SSL models.

**PLDA Hyperparameters:** For experiments where we directly used $wav2vec\_small$ (or XSLR) without fine-tuning for PLDA, we kept the original embedding dimension of 768 vector size (or 1024). For this setup, we observed that it is crucial to have enough training points for PLDA for each label – i.e.,

---

[10]For further fine-tuning of the MOS fine-tuned checkpoint on the main set on the OOD set we used learning rate 0.0001 and weight decay of 0.00001 which we estimated in informal experiments.

Table 1: *Results of MooseNet NN variants and PLDA. We report system-level MSE and SRCC on the VoiceMOS main and OOD track test sets, averaged over 10 runs, with standard deviations. See Figure 3 for core experiment setups; ablation experiments are shown in italics with the same prefix as the core experiment. We highlight the best results for each section (a group of similar experiments) in bold.*

| Main test system-level: | MSE | SRCC |
|---|---|---|
| **LDNet baseline** | 0.178 | 0.873 |
| **SSL-Baseline (B01)** | 0.148 | 0.921 |
| *W2V_main w/o contrast* | 0.149±0.033 | 0.922±0.007 |
| *W2V_main w/o augmnt.* | **0.137**±0.047 | 0.922±0.005 |
| *W2V_main w/o STOI* | 0.140±0.033 | 0.922±0.007 |
| *W2V_main_logCosh/Gauss* | 0.159±0.035 | 0.922±0.006 |
| W2V_main | 0.142±0.032 | **0.923**±0.006 |
| *W2V_main 50% train* | **0.150**±0.044 | **0.924**±0.006 |
| *W2V_main 5% train* | 0.307±0.176 | 0.884±0.006 |
| *W2V_main 136 train* | 0.289±0.072 | 0.853±0.006 |
| XSLR_main | 0.117±0.035 | 0.929±0.007 |
| W2V_main+PLDA_main | 0.105±0.009 | 0.922±0.006 |
| XSLR_main+PLDA_main | **0.101**±0.010 | **0.929**±0.005 |
| W2V+PLDA_main | 0.167±0.000 | **0.867**±0.000 |
| XLSR+PLDA_main | **0.076**±0.326 | 0.804±0.109 |

| OOD test system-level: | MSE | SRCC |
|---|---|---|
| **LDNet baseline** | 0.091 | 0.934 |
| **SSL-Baseline (B01)** | 0.099 | 0.975 |
| W2V_main | 2.657±0.399 | 0.710±0.040 |
| XLSR_main | 2.630±0.301 | 0.748±0.041 |
| W2V_main+PLDA_ood | **0.190**±0.061 | 0.860±0.042 |
| XLSR_main+PLDA_ood | 0.197±0.051 | **0.866**±0.039 |
| W2V_ood | 0.263±0.128 | 0.955±0.013 |
| XLSR_ood | **0.058**±0.011 | 0.942±0.007 |
| W2V_ood+PLDA_ood | 0.063±0.008 | **0.956**±0.011 |
| XLSR_ood+PLDA_ood | 0.062±0.008 | 0.945± 0.004 |
| W2V_solo-ood | 0.265±0.144 | 0.927±0.023 |
| W2V+PLDA_ood | **0.057**±0.009 | **0.955**±0.001 |
| XLSR+PLDA_ood | 0.145±0.012 | 0.886±0.018 |

have more than five MOS scores in each bin. See Figure 2(c).[11] We used whitening of input data,[12] we reduced the number of bins from 32 to 16 and used the first 64 PCA dimensions.

**Training Process:** The experiments run between 50 and 90 epochs on a single Nvidia A40. For the main track, the fine-tuning took between 60 and 100 minutes. For the OOD track, the fine-tuning runs for 20-40 minutes. We used the half-precision floats during all experiments. The PLDA training and inference use pre-computed embeddings, run on a single CPU for under two minutes for both tracks.

---

[11] https://github.com/oplatek/plda, a fork of https://github.com/RaviSoji/plda.

[12] We add Gaussian noise with 0.01 variance to NN embeddings.

# 6. Results

The results for both main and OOD tracks are summarized in Table 1; we refer back to research questions from Section 5 throughout the following text. Our MooseNet NN achieves better performance over the strong SSL baseline on the well-studied VoiceMOS challenge main track experiment *W2V_main* with respect to MSE and SRCC metrics. When the MooseNet NN checkpoint trained on the main track is fine-tuned to the OOD dataset, it matches the SSL baseline in terms of MSE in *W2V_ood* experiments. Interestingly, MooseNet PLDA improved those above-mentioned high-performing models in terms of MSE (see *W2V_main+PLDA_main* and *W2V_ood+PLDA_ood*).

The ablation study in the first six experiments in the table shows that the methods used for MooseNet NN training perform almost identically (RQ1). We observed that contrastive loss and STOI prediction multi-task fine-tuning favor SRCC ranking metric but produce inferior results for MSE. The difference was clearer for less-tuned models during the early stages of the training in our informal experiments.

As expected, the fine-tuned model on the main track, *W2V_main*, performs poorly on the OOD track (RQ2). Interestingly, PLDA trained on top this model's features improves performance both for MSE and SRCC by a large margin but does not match PLDA trained on top of OOD-tuned models (RQ3): PLDA consistently improves the *W2V_ood* and *XLSR_ood* models, which were first fine-tuned on the main and later on the OOD train sets. PLDA shines when used with non-finetuned neural models on the OOD track (RQ4). Refer to experiment *W2V+PLDA_ood*, which is our best model in terms of MSE and second best in terms of SRCC.

The MooseNet NN fine-tuning performs poorly if only the OOD set (of size 136 utterances) is used for training *W2V_solo-ood* models. The experiments with a reduced amount of training examples and *W2V_solo-ood* show that 136 training examples are enough for the SSL-finetuned models to reach performance close to LDNet [2] and with 249 examples it is possible to beat this baseline in terms of SRCC. With only 50% of training data, the *W2V_main 50% train* model achieves the performance of the strongest SSL baseline B01 (RQ5).

Finally, using the XLSR model instead of W2V seems to bring minor improvements on the main track but does not help on the OOD track. In such a constraint setting, the larger base model size loses its advantages (RQ6).

Note that our results averaged from ten runs are not directly comparable with the VoiceMOS challenge results. The submissions to the VoiceMOS scoreboard were selected based on the best performance on the development set. We observed that performance on the development set is well transferred to the test set performance. As a result, VoiceMOS Challenge results appear better, and the selection procedure favors models with higher variance, such as *W2V_solo-ood*.

# 7. Conclusion

We trained and thoroughly evaluated the trainable metrics MooseNet NN and MooseNet PLDA for Mean Opinion Score (MOS) prediction. We showed that SSL pre-trained models Wav2Vec-small and XLSR models learn useful representation for evaluating synthesized speech. Despite the wide adoption of the SSL model training for MOS prediction, we demonstrated that the SSL model fine-tuning could be easily improved, suggesting that the choice of data and training procedure is compli-

cated and likely an under-explored process. We experimented with the PLDA generative model, which is very fast to train, requires a minimum amount of data, and can be easily integrated on top of any fixed-size NN layer. We showed that PLDA consistently improves SSL speech models for MOS predictions task. For models tuned for MOS prediction but on the other dataset, PLDA improves both MSE and SRCC metrics by a large margin. When the PLDA is applied on top of fine-tuned, top-performing NN model, it tends to improve only the MSE metric. Interestingly, we showed that PLDA could be trained directly on top of an unmodified general-purpose SSL model on very small datasets (OOD) and still achieves performance comparable with LDNet VoiceMOS baseline [2].

## 8. Limitations and Future Work

Our work uses large SSL models which need a dedicated hardware accelerator for practical use. We hypothesize most prediction errors are due to inadequate training data on the extremes. We plan to develop techniques for both NN and PLDA models to predict epistemic uncertainty [29] in the future.

## 9. Acknowledgements

This research was supported by Charles University projects GAUK 40222 and SVV 260575, and by the European Research Council (Grant agreement No. 101039303 NG-NLG). It used resources provided by the LINDAT/CLARIAH-CZ Research Infrastructure (Czech Ministry of Education, Youth and Sports project No. LM2018101). The authors would like to thank the anonymous reviewers for their valuable feedback.

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
