# OpenReview forum: "MooseNet: A Trainable Metric for Synthesized Speech with a PLDA Module"
_Interspeech.org/2023/Workshop/SSW — SSW12_

### Official Review · Reviewer_Wpwp · 2023-05-29
**PLDA for MOS prediction**

**Rating:** 8
**Confidence:** 4

**Review:**

-Key Strength of the paper

This paper proposes the introduction of PLDA in the MOS prediction model, which transforms frame-by-frame acoustic features into time-invariant features by global pooling, an operation similar to that used to compute speaker vectors. While x-vector + PLDA is often used in speaker recognition, this paper evaluates a new approach, MOS prediction model + PLDA.

-Main Weakness of the paper

Since the speech synthesis field is unfamiliar with PLDA, more detailed descriptions suitable for researchers in the speech synthesis field are needed, such as the intuitive benefits of introducing PLDA in the MOS prediction task and how PLDA was adapted for multiple classes (5 classes of MOS).

-Novelty/Originality, taking into account the relevance of the work for the SSW audience

This paper is the first to use PLDA for MOS tasks, and it appears to be effective in predicting MOS for OOD tasks. This task is, of course relevant to speech synthesis researchers since it is a MOS value prediction for speech synthesis systems.

-Technical Correctness, is the work technically and/or scientifically solid? Are sufficient details provided to allow any experiments to be reproduced or equivalent experiments run?

Overall, the details are well described. Only the part about making PLDA compatible with multiple classes was not understood and needs to be added.

-Suggestions for improvement

There are many derived models in the experiment and the system identification names (e.g. w2v_main + PLDA_ood) are a bit confusing. If possible, it would be better to use a name that is easier to understand.

-Quality of References, is it a good mix of older and newer papers? Do the authors show a good grasp of the current state of the literature? Do they also cite other papers apart from their own work?

Yes.

-Clarity of Presentation, the English does not need to be flawless, but the text should be understandable

The content can be understood without problems.

---

> ### Author Response · Authors · 2023-06-28
> **Thank you**
>
> Thank you for spot-on feedback.

---

### Official Review · Reviewer_BWfr · 2023-06-03
**A MOS prediction paper with worthwhile new results that sometimes sound like they are better than they are**

**Rating:** 7
**Confidence:** 3

**Review:**

* Key Strength of the paper

The extensive experiments and the large number of comparisons made. The multiple runs made and reported are arguably an improvement over the methodology used in the VoiceMOS Challenge.

* Main Weakness of the paper

The paper fails to leverage a core strength of using open challenge data, which is that one may compare a proposed method to many systems, and not only a handful of baselines.

* Novelty/Originality, taking into account the relevance of the work for the SSW audience

The work is definitely relevant to the audience. The novelty is sufficient for publication.

* Technical Correctness, is the work technically and/or scientifically solid? Are sufficient details provided to allow any experiments to be reproduced or equivalent experiments run?

The text provides a good amount of detail for replicating the work. The increased page count of SSW over Interspeech is beneficial in this respect, since fewer details would fit on the four pages of content Interspeech allows.

* Quality of References, is it a good mix of older and newer papers? Do the authors show a good grasp of the current state of the literature? Do they also cite other papers apart from their own work?

This reviewer is not an expert in the MOS prediction literature, but there are seemingly no references to non-intrusive quality prediction prior to the FAD. There has been quite some work in learnt prediction of MOS test scores before (although generally with inferior results compared to the approaches of today), consider, e.g., the body of work due to Florian Hinterleitner, and other papers in the same area.

* Clarity of Presentation, the English does not need to be flawless, but the text should be understandable

This is a very carefully written paper, significantly above the standard of many (most?) published papers in speech synthesis. In fact, the writing is sometimes too carefully crafted, in that it is technically correct but tends to suggest interpretations that are more favourable to the proposed method than the results support.
Figures well complement the exposition.

--------------------

Detailed review comments, mostly ordered by the location in the paper that they apply to:

Comment that does not apply to a specific location in the paper: The bold subheadings differ in their capitalisation conventions (title vs. sentence case), and whether they are followed by a colon or not.

Comment that does not apply to a specific location in the paper: It seems like the MooseNet model has relatively few added parameters. It would be interesting to see a comparison of the model size of different variants compared to the two baselines (and possibly to other systems from the VoiceMOS Challenge as well), for instance in the form of another table.

Page 1, column 1, abstract: "The fine-tuned MooseNet NN with the PLDA module achieves the best results, surpassing strong baselines on the VoiceMOS Challenge data." This is technically correct but likely to mislead the reader somewhat. The proposed method surpasses the SSL baseline in most but not all respects, which should be articulated. And "baselines" really means the designated baselines from the VoiceMOS Challenge evaluation, not the (entirely plausible) alternative reading that strong models submitted for the VoiceMOS Challenge are being used as baselines for this work. These nuances not being articulated, a straightforward reading of the abstract overstates the claims compared to what the empirical results can support.

Page 2, column 1, Sec. 2: "Trainable neural metrics directly predicting MOS dominate in automatic speech evaluation as they show a high correlation with human evaluation, not only on the system level but also on the utterance level" This statement (especially the part about utterance-level results) should be backed up with one or more relevant citations, such as the VoiceMOS paper.

Page 2, column 2, Fig. 2: "Moosenet" should be "MooseNet"

Page 2, column 2, Sec. 3: "Gauss loss stabilizes and improves the training which we explain as better modeling the MOS variance." The remark on stability is surprising since fitting Gaussians using maximum-likelihood estimation is widely considered less numerically stable than plain MSE (being equivalent to Gaussian MLE with a fixed standard deviation), since small estimated variances can lead to a division by zero. It is possible that the use of gradient clipping (described later in the paper) contributed towards the claimed stability. Consider nuancing this statement.

Page 3, column 1, Sec. 3: "Saeki et al.[7]" should be "Saeki et al. [7]"

Page 3, column 1, footnote 5 is missing a full stop at the end of the sentence.

Page 3, column 3, Sec. 5: There is a nice set of research questions enumerated here. Should the submission be accepted, the authors could consider spending some of the extra page allotted to accepted papers to explicitly refer to and discuss the enumerated points in Sec. 6. As it is, many of them are covered in the discussion, but in another order than here and with no references to the numbers.

Page 4, Fig. 3: This is a really illustrative figure.

Page 4, column 2, Sec. 5: "where we used directly" would perhaps be better as "where we directly used"

Page 4, column 2, Sec. 5: "embedding dimension o 768" should probably be "embedding dimension of 768"

Page 5, column 1, Table 1: Although the bolding in the table is consistent, it can be misleading, since the reader has to look carefully to notice that the bolded results sometimes signify a performance level that is below the SSL baseline. Consider addressing this with a comment, to avoid incorrect readings.

Page 5, column 1, Sec. 6: "the strong SSL baseline" the text should explicitly confirm whether or not this is the same system called B01 ("SSL-MOS") in the VoiceMOS paper.

Page 5, column 2, Sec. 6: "As a result, VoiceMOS Challenge results appear better" This reviewer really appreciated this entire paragraph. However, it is nonetheless recommended to expand on the discussion/comparison to the VoiceMOS Challenge and its results. The true strength of that challenge is not in the baselines, but that so many different performance numbers are available that are comparable to each other. How does the performance numbers of MooseNet stack up against the other systems in the challenge, knowing that this is an imperfect comparison? Many submitted systems outperformed the SSL baseline B01 in terms of SRCC on the main task, but not the OOD task. Even better, is it possible to select a MooseNet system "based on the best performance on the development set", whose performance then can be compared directly to the wealth of numbers in the VoiceMOS Challenge? What happens then?

Page 5, column 2, Sec. 6: There is no discussion of the statistical significance of the improvements achieved. That might be acceptable, since it is not clear if significance is achievable in this setup. However, the paper could benefit from the discussion being even more mindful of the effects of stochastic variation in the results. This could involve looking more at the different variances seen across runs, but especially an empirically-supported argument (and possibly analysis) that tries to identify how statistically robust the conclusions drawn are to stochastic factors.

Page 5, column 1, Sec. 6: "The fine-tuning to the OOD dataset of the MooseNet NN checkpoint trained on the main track achieves does match the SSL base- line especially in terms of MSE in W2V_ood experiments." The use of "especially" here suggests a reading that the proposed method matched the SSL baseline in all respects, but was especially good for MSE. That is not accurate, since it did not come close to matching the SSL baseline in terms of SRCC. The text should be explicit that the SSL baseline achieved notably better SRCC.

Page 5, column 2, Sec. 6: One instance of the word "PLDA" is bolded. The reason for doing so is not clear to this reader.

Page 6, column 1, Sec. 9: References [4] and [17] are both NeurIPS papers, but they use very different ways to refer to that venue ("NeurIPS" vs. "Advances in neural information processing systems"). One has publisher information whereas the other does not.

Page 6, column 1, Sec. 9: Reference [6] uses inconsistent, sentence-case capitalisation, with MOS written as "MOS". Reference [13] has a similar issue.

Page 6, column 1, Sec. 9: Reference [11], unlike most other Interspeech paper, mentions ISCA in the reference information.

Page 6, column 1, Sec. 9: Reference [15] is missing a publication venue. Perhaps specify that it's an arXiv preprint (if that's the case) and give the unique preprint number, or otherwise indicate where the work can be found.

---

> ### Author Response · Authors · 2023-06-28
> **Thank you**
>
> Thank you very much for the detailed feedback.

---

### Decision · Program_Chairs · 2023-06-14

**Decision:**

Accept

**Comment:**

SSW2003 received 45 papers. The acceptance rate is 82%. We are pleased to inform you that your paper has been accepted by the SSW2023 Program Committee. Please read the reviews carefully and submit your camera-ready paper by June 28th. Most reviewers performed a detailed review. Please answer to their questions and consider their comments. Note that camera-ready papers are credited with one extra page to allow authors to consider reviewers’ suggestions. So max 7 pages in total including figures & refs.
The deadline for submitting the revised version (with full non-anonymized authors and refs!) is 28th June.